# Brief communication: Impact of the recent atmospheric circulation change in summer on the future surface mass balance of the Greenland ice sheet

Alison Delhasse [1], Xavier Fettweis [1], Christoph Kittel [1], Charles Amory [1], and Cécile Agosta [1,2]

[1]Laboratory of Climatology, Department of Geography, University of Liège, Liège, Belgium
[2]Laboratoire des Sciences du Climat et de l'Environnement, Gif-sur-Yvette, France

*Correspondence to:* Alison Delhasse (alison.delhasse@uliege.be)

**Abstract.** Since the 2000's, a change in the atmospheric circulation over North Atlantic resulting in more frequent blocking events has favoured warmer and sunnier weather conditions over the Greenland Ice sheet (GrIS) in summer enhancing the melt increase. This circulation change is not represented by General Circulation Models (GCMs) of the 5th Coupled Model Intercomparison Project (CMIP5) which do not predict any circulation change for the next century over the North Atlantic. The goal of this study is to evaluate the impact of an atmospheric circulation change (as currently observed) on projections of the future GrIS surface mass balance (SMB). We compare GrIS SMB estimates simulated by the regional climate model MAR forced by perturbed reanalysis (ERA-Interim with a temperature correction of +1 °C, +1.5 °C and +2 °C at the MAR lateral boundaries) over 1980 – 2016 to projections of the future GrIS SMB from MAR simulations forced by three GCMs over selected periods for which a similar temperature increase of +1 °C, +1.5 °C and +2 °C is projected by the GCMs in comparison to 1980 – 1999. Mean SMB anomalies produced with perturbed reanalysis over the climatologically stable period 1980 – 1999 are similar to those produced with MAR forced by GCMs over future periods characterised by a similar warming over Greenland. However, over the two last decades (2000 – 2016) when an increase in the frequency of blocking events has been observed in summer, MAR forced by perturbed reanalysis suggests that the SMB decrease could be amplified by a factor of two if such atmospheric conditions persist compared to projections forced by GCMs for the same temperature increase but without any circulation change.

## 1 Introduction

Starting in the late 1990's, the surface mass balance (SMB, i.e. the difference between mass sources and sinks at the surface) of the Greenland Ice Sheet (GrIS) has been decreasing through a rise in surface meltwater runoff (Fettweis et al., 2013a; van den Broeke et al., 2016). Since the 2000's, multiple melting records have been broken over the GrIS (Tedesco et al., 2013; Hanna et al., 2014; Fettweis et al., 2017) and 70% of this melt increase can be attributed to an atmospheric circulation change gauged through a shift of the summer North Atlantic Oscillation (NAO) index to negative values (Fettweis et al., 2013b). These changes have favoured anticyclonic conditions (i.e. more frequent blocking events) in summer over Greenland which enhance the melt-albedo feedback through three main ways (Box et al., 2012): (1) an increase in advection of warm air along the western coast

of the ice sheet enhancing the surface sensible heat flux, thus strengthening snow grain metamorphism and further decreasing surface albedo (Tedesco et al., 2016); (2) a decrease in snowfall rates in summer leading to longer exposure of bare ice at the GrIS margins (Noël et al., 2015) ; and (3) an increase in incoming radiation resulting in more surface heating. For this last feature, the debate about the contribution of shortwave and longwave radiations is still ongoing as the role of clouds is opposite

between the ablation zone (more sensitive to shortwave anomalies) and the accumulation zone (more sensitive to longwave anomalies). Hofer et al. (2017) points out the importance of increasing incoming solar radiation while other studies allocate the increase in surface heating and melting to longwave radiation notably due to the transport of water vapor (Neff et al., 2014; Bonne et al., 2015) and liquid low-level clouds (Bennartz et al., 2013). These processes reduce meltwater refreezing and then enhance meltwater runoff (Van Tricht et al., 2016).

Such an amplification in surface melt is well represented by Regional Climate Models (RCMs) forced by climate reanalysis which capture the current circulation change (Ettema et al., 2009; Fettweis et al., 2011, 2017; Noël et al., 2015, 2018). However, as General Circulation Models (GCMs) do not presently predict any future circulation change (Belleflamme et al., 2012; Fettweis et al., 2013b; Hanna et al., in review, 2018), the melt increase currently observed is underestimated when RCMs are forced by GCM scenarios starting from 2000's (e.g., Fettweis et al., 2011, 2013b; Rae et al., 2012). This raises the question of

how RCM-based projections of future GrIS SMB are affected by the GCM forcing if the recent shift to negative NAO phases in summer persists through the next decades?

To address this issue, we use the Modèle Atmosphérique Régional (MAR), especially developed for polar regions, to perform a sensitivity study based on the analysis of MAR-derived GrIS SMB anomalies resulting from various forcing experiments with ERA-Interim reanalysis and three GCMs from the CMIP5 database (The 5th phase of the Climate Model Intercomparison

Project; Taylor et al., 2012).

## 2   Data and methodology

### 2.1   The regional climate model MAR

The model MAR is a RCM specifically developed for simulating polar climate specificities (e.g., Amory et al., 2015; Gallée et al., 2015; Lang et al., 2015), furthermore abundantly evaluated over the Greenland ice sheet (e.g., Fettweis et al., 2011,

2017). In this study, we use the version 3.8 of MAR, and refer to Fettweis et al. (2017) for a more detailed description of MAR. Relative to the version 3.5 used in Fettweis et al. (2017) and in addition to usual bug fixes and improved computation efficiency, the main improvement of MAR v3.8 consists of an increase of the cloud life time, partly correcting the underestimation of downward longwave radiation and the overestimation of inland precipitation described in Fettweis et al. (2017).

MAR requires prescription of atmospheric fields (temperature, specific humidity, wind speed, pressure) at its lateral bound-

aries as well as sea surface conditions (SSC, defined as sea ice concentration and sea surface temperature) from climate reanalysis or GCM outputs. Using ERA-Interim reanalysis (Dee et al., 2011) to force MAR, Fettweis et al. (2013a) have shown that the period 1980 – 1999 is characterised by a stable climate and general circulation over Greenland, before a circulation shift

**Table 1.** 20-yr periods corresponding to an increase in temperature of ∼+1 °C, ∼+1.5 °C and ∼+2 °C for the three selected GCMs (MIROC5, NorESM1 and CanESM2). The exact increase in temperature and the standard deviation for these periods compared with 1980 – 1999 are shown.

|  | MIROC5 | | NorESM1 | | Can ESM2 | |
|---|---|---|---|---|---|---|
|  | Temp. increase (°C) | Period | Temp. increase (°C) | Period | Temp. increase (°C) | Period |
| ∼+0 °C | 0.00±0.54 | 1980 – 1999 | 0.00±0.41 | 1980 – 1999 | 0.00±0.57 | 1980 – 1999 |
| ∼+1 °C | 1.00±0.39 | 2007 – 2027 | 0.99±0.52 | 2014 – 2034 | 1.00±0.41 | 1997 – 2017 |
| ∼+1.5 °C | 1.52±0.52 | 2019 – 2039 | 1.51±0.57 | 2023 – 2043 | 1.49±0.60 | 2006 – 2026 |
| ∼+2 °C | 1.97±0.64 | 2029 – 2049 | 2.00±0.55 | 2033 – 2053 | 1.99±0.49 | 2016 – 2036 |

occurred in summer at the end of the 1990's. In this study, we thus consider the period 1980 – 1999 as a reference (Fettweis et al., 2013a), and discuss mean GrIS annual SMB anomalies with respect to this reference period.

## 2.2 Forcing experiments

We performed two sets of sensitivity experiments according to the large-scale forcing used, as described below.

### 2.2.1 ERA-Interim forcing

MAR was first forced by the ERA-Interim reanalysis every 6 hours at a spatial resolution of 25 km over 1979 – 2016. Two distinct periods are considered: the reference period 1980 – 1999, and the period 2000 – 2016 for which a different atmospheric circulation has been observed in summer on average by comparison with the reference period (Fettweis et al., 2013b, 2017; Hanna et al., in review, 2018). Then, we performed sensitivity experiments in which ERA-Interim atmospheric temperatures were increased by respectively +1 °C, +1.5 °C and +2 °C at each of the 24 vertical sigma levels of the MAR atmospheric lateral boundaries (hereafter referred to as perturbed reanalysis). The mean SMB anomalies of these sensitivity experiments are referred to as MARera+1, MARera+1.5 and MARera+2 respectively for anomalies averaged over 1980 – 1999, and MARera2k, MARera2k+1, MARera2k+1.5 and MARera2k+2 for corresponding anomalies averaged over 2000 – 2016 (Appendix A, Table A1). Note that the relative humidity at the lateral boundaries was conserved by modifying the specific humidity according to temperature changes in order to obtain precipitation fields consistent with warmer ERA-Interim atmospheric temperatures. However, SSC from ERA-Interim reanalysis remained unchanged in the sensitivity experiments.

### 2.2.2 GCM forcing

We performed three additional experiments in which MAR was forced over 1980 – 2060 with three GCMs from the CMIP5 database, using the Historical scenario over 1980 – 2005 and the future RCP4.5 scenario (Moss et al., 2010) over 2005 – 2060. Then, for each GCM, three 20-yr periods (different for each GCM) characterised by a climate of about ∼+1 °C, ∼+1.5 °C

and ∼+2 °C warmer on average than the climate over the reference period (as represented by the GCM) of 1980 – 1999 were considered (Table 1). The differences between the GCM periods are only due to the timing required to reach each relative warming. Note that each warming was computed as the mean JJA (June-July-August) temperature anomaly compared to 1980 – 1999 over the computation domain (100°W- 0°W, 55°N-85°N) at four vertical levels in the free atmosphere (850 hPa, 700 hPa, 600 hPa and 500 hPa).

The three GCMs used are CanESM2, NorESM1 and MIROC5, identified in Fettweis et al. (2013b) as the CMIP5 GCMs best representing the general circulation at 500 hPa (influencing the precipitation amount and pattern simulated by MAR) and the JJA temperature at 700 hPa over Greenland (influencing the melt amount simulated by MAR) compared to ERA-Interim over 1980 – 1999. However, some discrepancies remain between MAR forced by these GCMs and MAR forced by ERA-Interim over 1980 – 1999. For instance, MAR underestimates runoff and overestimates snowfall when forced by NorESM1 compared to MAR forced by ERA-Interim (Fettweis et al., 2013a). This is also confirmed by a recent study comparing melt energy budgets from MAR forced by ERA-Interim, MIROC5, NorESM1 and CanESM2 (Leeson et al., 2018). This is why the SMB obtained with perturbed reanalysis cannot be directly compared to the SMB obtained with GCM forcing data because of these divergences over the reference period.

Therefore we compare anomalies of these GCM-forced projections during the 20-yr future periods of ∼+1 °C, ∼+1.5 °C and ∼+2 °C to the corresponding GCM-forced MAR climate over the reference period (1980 – 1999) and not directly to MAR forced by ERA-Interim over 1980 – 1999. As for the sensitivity experiments using the ERA-Interim reanalysis as forcing, the mean SMB anomalies associated with these GCMs experiments are respectively referred to as MARcan+x for CanESM2, MARmir+x for MIROC5 and MARnor+x for NorESM1 where x equals 1, 1.5 or 2 corresponds to a warming of respectively +1 °C, +1.5 °C and +2 °C (Appendix A, Table A2). Contrary to ERA-Interim forced experiments, no humidity correction has been applied at the MAR lateral boundaries and the SSC are directly prescribed from the RCP4.5 scenario projected by the respective GCM.

# 3   Results

## 3.1   Analysis of warming experiments without circulation change

Before assessing the impact of the current circulation change on SMB projections, we evaluate analogies between MAR forced by perturbed reanalysis and by GCM scenarios over future periods experiencing a similar warmer climate. Figure 1 shows difference in SMB anomalies between MARera+1 (resp. +2) and MARmir+1 (resp. +2, using MIROC5 as forcing) over 1980 – 1999 when the mean general circulation in summer is similar in both ERA-Interim and GCMs.

Even if the differences are mainly not statistically significant (i.e. lower than the inter-annual variability of the simulation of MAR forced by unaltered ERA-Interim over 1980 – 1999), the runoff anomalies are generally higher if MAR is forced by GCMs rather than by corresponding perturbed reanalysis. GCM-forced simulations also predict a higher precipitation increase along the south-east of the ice sheet. These weak differences are caused by the SSC which were not modified in experiments based on perturbed reanalysis while the GCM-forced simulations use future SSC. Anomalies at the ice sheet margins (Fig. 1)

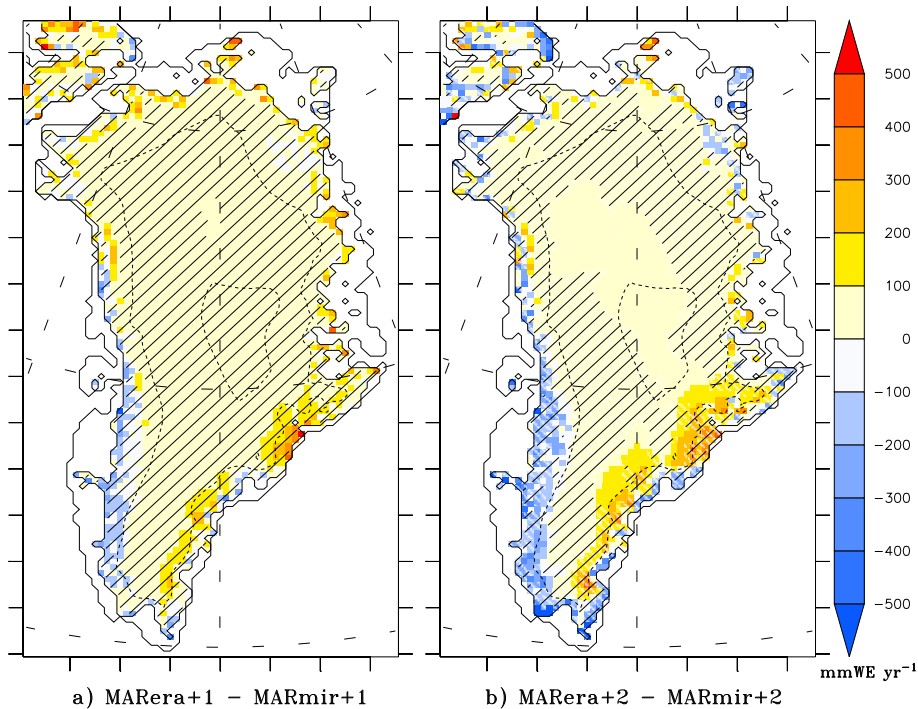

a) MARera+1 − MARmir+1          b) MARera+2 − MARmir+2

**Figure 1.** Differences of mean anomalies of annual SMB (in mmWE $y^{-1}$) between a) MARera+1 and MARmir+1 and b) MARera+2 and MARmir+2. Areas where anomaly differences are smaller than the inter-annual variability (i.e. the standard deviation) of the simulation of MAR forced by unaltered ERA-Interim over 1980 – 1999 are hatched. Dashed lines are equal altitude lines of 2000 m and 3000 m. See Appendix A Table A1 and Table A2 for abbreviations.

are similar to the SMB anomalies found by Noël et al. (2014) who evaluated the (insignificant) impacts of SSC on the SMB of the Greenland ice sheet. We therefore assume that anomalies between MARera+x and MARmir (or MARnor or MARcan) result from the SSC unchanged in MARera+x. Because GCMs fail to represent the circulation change observed in summer over Greenland since 2000's, MAR forced by perturbed reanalysis over 1980 – 1999 simulates similar SMB anomalies as MAR forced by GCMs over the corresponding future warmer periods. Therefore, evaluating MAR forced by perturbed reanalysis over 2000 – 2016 allows us to evaluate the likely impact of a warmer climate induced by an increase of blocking events (as currently observed) on the projected GrIS SMB.

## 3.2 Influence of a potential future circulation change

Over 2000 – 2016, the JJA temperature in the ERA-Interim reanalysis has increased by 0.7 °C in the free atmosphere (mean 850-500 hPa temperature integrated over the computation domain) compared to 1980 – 1999. This offset in temperature has thus to be taken into account when comparing the anomalies of the different warming experiments. The SMB anomaly MARera2k+1 (resp. MARera2k+1.5) is significantly more negative (up to a factor of two) than perturbed-reanalysis experiments

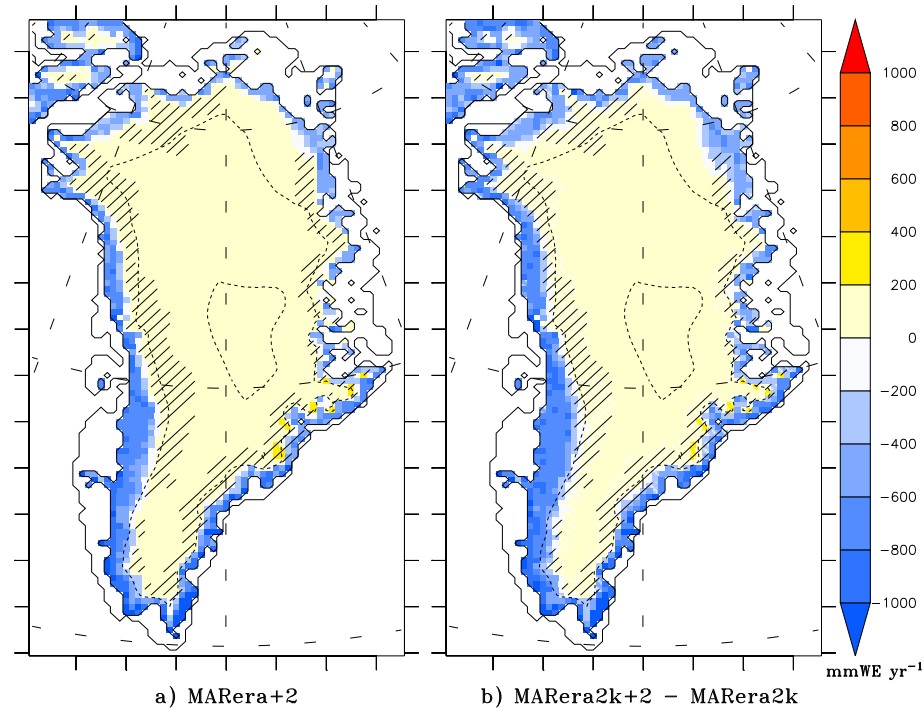

a) MARera+2          b) MARera2k+2 − MARera2k

**Figure 2.** a) Mean anomalies of annual SMB (mmWE y$^{-1}$) of MAR forced by warmer ERA-Interim warmer of +2 °C compared to MAR forced by unaltered ERA-Interim over 1980 – 1999. b) Differences between mean anomalies of annual SMB (mmWE y$^{-1}$): MARera2k+2 - MARera2k. Areas where anomaly differences are smaller than the inter-annual variability (i.e. the standard deviation) of the simulation of MAR forced by unaltered ERA-Interim over 1980 – 1999 are hatched. Dashed lines are equal altitude lines of 2000 m and 3000 m. See Appendix A Table A1 and Table A2 for abbreviations.

and GCM-forced future experiments relative to a climate warmer by +1.5 °C (resp. +2 °C) (Table 2). This suggests that capturing the recent circulation change simulated by perturbed-reanalysis experiments would enhance the projected SMB decrease compared to the decrease projected for a warmer climate without any circulation change. This is illustrated in Figure 2 where the decrease in SMB is amplified for an equal increase in temperature of +2 °C over 2000 – 2016 (Fig. 2b), i.e. including
5   the recent circulation change, compared to the decrease for the reference circulation over 1980 – 1999 (Fig. 2a). Additional 2D-representations of these differences for GCM experiments are available in supplementary materials.

As runoff (RU) and snowfall (SF) mainly drive the GrIS SMB (Box et al., 2004), we mainly discuss in the following the anomalies relative to these two components. As for SMB anomalies, RU and SF anomalies are computed as differences between the corresponding mean value for a given experiment and the mean value for the reference period using the unaltered large-
10  scale forcing (Table 2). Even though non-significant, an increase in SF is observed for all experiments associated with a rise in temperature in response to a higher air capacity for holding water vapor (Fettweis et al., 2013a). Moreover, mean RU anomalies increase with the temperature rise in all warming experiments, most significantly for the experiments using perturbed reanalysis

**Table 2.** Mean GrIS integrated anomalies of annual SMB (Gt y$^{-1}$), runoff (RU), snowfall (SF) and melt (ME) compared to 1980 – 1999. Anomalies from GCM-forced simulations are given as averaged. Anomalies greater than the 1980 – 1999 standard deviation (i.e. greater than the inter-annual variability) of the simulation of MAR forced by unaltered ERA-Interim are shown in bold.

| | | | Forcing | |
| --- | --- | --- | --- | --- |
| | Temperature increase (°C) | ERA-Interim 1980 – 1999 (MARera) | ERA-Interim 2000 – 2016 (MARera2k) | Mean of the 3 GCMs |
| Annual mean SMB (Gtyr$^{-1}$) | +0 | 0 | **-205** | 0 |
| | +1 | -84 | **-326** | -118 |
| | +1.5 | -146 | **-408** | -164 |
| | +2 | **-206** | **-492** | **-197** |
| Annual mean RU (Gtyr$^{-1}$) | +0 | 0 | **-211** | 0 |
| | +1 | **142** | **393** | **141** |
| | +1.5 | **236** | **508** | **215** |
| | +2 | **328** | **626** | **283** |
| Annual mean SF (Gtyr$^{-1}$) | +0 | 0 | -8 | 0 |
| | +1 | 37 | 28 | 13 |
| | +1.5 | 56 | 46 | 29 |
| | +2 | **75** | **63** | 51 |
| Annual mean ME (Gtyr$^{-1}$) | +0 | 0 | 195 | 0 |
| | +1 | **133** | **352** | **135** |
| | +1.5 | **210** | **440** | **203** |
| | +2 | **291** | **534** | **261** |

over 2000 – 2016 when the circulation change has occurred. It can thus be concluded that runoff is mainly responsible for the SMB discrepancies between the different sensitivity experiments in a warmer climate. As for RU, melt (ME) anomalies are also amplified with the circulation change (Table 2). However, RU anomalies are systematically higher than ME anomalies. This can be explained by two factors (Machguth et al., 2016): (1) there is less pore place available for meltwater storage in warmer and then more saturated firn and, (2) the bare ice area (in the ablation zone) is larger in warmer climate reducing the potential surface for meltwater storage which also amplifies the meltwater runoff increase. The future decrease of the ice sheet meltwater retention capacity has notably been discussed by Van Angelen et al. (2013).

On average over the ice sheet, downward shortwave radiation (SWD) and downward longwave radiation (LWD) are respectively almost 4 Wm$^{-2}$ and 3 Wm$^{-2}$ higher over the 2000 – 2016 period than over the reference period (Table 3). This concords with Hofer et al. (2017)'s conclusions which argue that the current observed melt increase since the 2000's is a combination

**Table 3.** Mean GrIS integrated anomalies of summer energy fluxes (W.m$^{-2}$) and summer surface 2-m temperature (°C) compared to 1980 – 1999. Anomalies from GCM-forced simulations are given as averaged. Anomalies greater than the 1980 – 1999 standard deviation (i.e. greater than the inter-annual variability) of the simulation of MAR forced by unaltered ERA-Interim are shown in bold.

| | | Forcing | | |
| --- | --- | --- | --- | --- |
| | Temperature increase (°C) | ERA-Interim 1980 – 1999 (MARera) | ERA-Interim 2000 – 2016 (MARera2k) | Mean of the 3 GCMs |
| | +0 | 0.0 | **3.7** | 0.0 |
| JJA mean | +1 | -2.7 | 0.9 | -0.7 |
| SWD (Wm$^2$) | +1.5 | **-4.2** | -0.6 | -2.4 |
| | +2 | **-5.8** | -2.2 | -3.9 |
| | +0 | 0.0 | **3.2** | 0.0 |
| JJA mean | +1 | **4.8** | **8.1** | **4.5** |
| LWD (Wm$^2$) | +1.5 | **7.2** | **10.6** | **7.3** |
| | +2 | **9.7** | **13.2** | **9.4** |
| | +0 | 0.0 | **5.4** | 0.0 |
| JJA mean absorbed | +1 | 1.9 | **7.5** | 2.0 |
| SWD (Wm$^2$) | +1.5 | **3.0** | **8.8** | 2.8 |
| | +2 | **4.0** | **10.0** | 3.9 |
| | +0 | 0.00 | **1.24** | 0.00 |
| JJA mean | +1 | **0.97** | **2.20** | **1.07** |
| T2m (°C) | +1.5 | **1.45** | **2.68** | **1.62** |
| | +2 | **1.93** | **3.15** | **2.01** |

of the increase in SWD and LWD. However both simulations forced by perturbed reanalysis as well as in GCM-forced simulations with a warmer climate suggest a strong SWD decrease as a result of an increasing cloud cover, as already discussed in Franco et al. (2013). This effect combined with a higher free atmosphere temperature explains then the increase in LWD in a warmer climate (Hofer et al., 2017).

5   Due to the enhanced positive melt-albedo feedback since the 2000's, SWD absorbed by the surface is more than two times higher in simulations with perturbed reanalysis over 2000 – 2016 than over the reference period. Due to a lower albedo (in particular in the ablation zone), the surface absorbs more energy, amplifying the melt increase which further decreases the albedo, potentially reaccelerating melting in addition to a decrease of the ice sheet capacity to refreeze meltwater. This positive feedback triggered by more frequent anticyclonic summer situations over Greenland causes a runoff increase nearly two times

10   higher in simulations over 2000 – 2016 than in the simulations over the reference period.

In the absence of a circulation change, the increase in near-surface temperature (T2m) is only due to the temperature increase prescribed at the MAR lateral boundaries (Table 3). With a circulation change, the increase in T2m is higher and no more uniform as a result of enhanced warm air advection along the West GrIS (see Fig. S1 in supplementary material). Such more frequent anticyclonic conditions, resulting from more frequent blocking events (Hanna et al., in review, 2018), explain the increase in cloud cover at the north of the ice sheet and the associated LWD increase shown in Hofer et al. (2017) in this area, while sunnier conditions dominate in the southern part.

## 4 Conclusions

The goal of this study is to assess the impact of unresolved recent atmospheric circulation change in GCMs on RCM-based projections of future GrIS SMB. For this purpose, we used the RCM MAR and performed forcing sensitivity experiments with the ERA-Interim reanalysis and large-scale fields from 3 selected GCMs of the CMIP5 database to investigate the influence of a warmer atmosphere in the context of a circulation change inducing more frequent blocking events.

We used the annual SMB produced with each original forcing over the climatologically stable period of 1980 – 1999 as a reference, to compute mean annual SMB anomalies relative to each forcing experiment.

The first experiments consisted of increasing the atmospheric temperature in the ERA-Interim reanalysis by +1 °C, +1.5 °C and + 2 °C at the MAR lateral boundaries over two distinct periods, i.e. 1980 – 1999 and 2000 – 2016, respectively before and after the shift in the summer NAO index. Additional forcing experiments were then performed using the three selected GCMs to force MAR over the reference period and over 20-yr future periods for which a similar temperature increase of ∼+1 °C, ∼+1.5 °C and ∼+2 °C is predicted by these GCMs in the free atmosphere.

The comparison between SMB anomalies relative to perturbed reanalysis and GCM future experiments revealed similar results for each corresponding warming experiment, since for each GCM the atmospheric circulation remains unchanged over time. This allowed us to evaluate the likely impact of a warmer climate induced by a circulation change on the GrIS SMB by comparing SMB anomalies relative to perturbed reanalysis over 2000 – 2016 with GCM-forced future SMB anomalies. This comparison suggests that capturing the circulation change leads to SMB anomalies two times higher on average with a circulation change than without for a similar atmospheric warming. These higher anomalies are explained by more frequent summer anticyclonic situations over GrIS leading to an increase in SWD and warm air advection along the west coast, both promoting a decrease in albedo. As a result, the runoff increase is enhanced and is responsible for the higher decrease in SMB.

The results of this study suggest that previous estimates of GrIS melt projections produced using CMIP5 data as forcing (e.g., Rae et al., 2012; Fettweis et al., 2013a) could be significantly underestimated if the current summer atmospheric circulation pattern over Greenland persists. In the context of the forthcoming 6th phase of the Coupled Model Intercomparison Project, our conclusions highlight the importance of examining whether GCMs predict circulation changes in the next decades and, if so, of evaluating their potential influence on projections of the future GrIS SMB. Another remaining scientific challenge of particular interest for GrIS SMB projections is to establish if the increasing frequency of blocking events could be linked to global climate change.

# Appendix A:  Description of the abbreviations used in this study

**Table A1.** Abbreviation description of reanalysis sensitivity experiments

| | |
|---|---|
| MARera+1 | Anomalies between MAR forced by the ERA-Interim reanalysis warmer of +1 °C over 1980 – 1999 and MAR forced by the unaltered ERA-Interim reanalysis over 1980 – 1999 |
| MARera+1.5 | Anomalies between MAR forced by the ERA-Interim reanalysis warmer of +1.5 °C over 1980 – 1999 and MAR forced by the unaltered ERA-Interim reanalysis over 1980 – 1999 |
| MARera+2 | Anomalies between MAR forced by the ERA-Interim reanalysis warmer of +2 °C over 1980 – 1999 and MAR forced by the unaltered ERA-Interim reanalysis over 1980 – 1999 |
| MARera2k | Anomalies between MAR forced by the ERA-Interim over 2000 – 2016 and MAR forced by the unaltered ERA-Interim reanalysis over 1980 – 1999 |
| MARera2k+1 | Anomalies between MAR forced by the ERA-Interim reanalysis warmer of +1 °C over 2000 – 2016 and MAR forced by the unaltered ERA-Interim reanalysis over 1980 – 1999 |
| MARera2k+1.5 | Anomalies between MAR forced by the ERA-Interim reanalysis warmer of +1.5 °C over 2000 – 2016 and MAR forced by the unaltered ERA-Interim reanalysis over 1980 – 1999 |
| MARera2k+2 | Anomalies between MAR forced by the ERA-Interim reanalysis warmer of +2 °C over 2000 – 2016 and MAR forced by the unaltered ERA-Interim reanalysis over 1980 – 1999 |

**Table A2.** Abbreviation description of GCM sensitivity experiments

| | |
|---|---|
| MARmir+1 | Anomalies between MAR forced by MIROC5 over a warmer 20-yr period of +1 °C relatively to the reference period 1980 – 1999 and MAR forced by MIROC5 over the reference period |
| MARmir+1.5 | Anomalies between MAR forced by MIROC5 over a warmer 20-yr period of +1.5 °C relatively to the reference period 1980 – 1999 and MAR forced by MIROC5 over the reference period |
| MARmir+2 | Anomalies between MAR forced by MIROC5 over a warmer 20-yr period of +2 °C relatively to the reference period 1980 – 1999 and MAR forced by MIROC5 over the reference period |
| MARnor+1 | Anomalies between MAR forced by NorESM1 over a warmer 20-yr period of +1 °C relatively to the reference period 1980 – 1999 and MAR forced by MIROC5 over the reference period |
| MARnor+1.5 | Anomalies between MAR forced by NorESM1 over a warmer 20-yr period of +1.5 °C relatively to the reference period 1980 – 1999 and MAR forced by MIROC5 over the reference period |
| MARnor+2 | Anomalies between MAR forced by NorESM1 over a warmer 20-yr period of +2 °C relatively to the reference period 1980 – 1999 and MAR forced by MIROC5 over the reference period |
| MARcan+1 | Anomalies between MAR forced by CanESM2 over a warmer 20-yr period of +1 °C relatively to the reference period 1980 – 1999 and MAR forced by MIROC5 over the reference period |
| MARcan+1.5 | Anomalies between MAR forced by CanESM2 over a warmer 20-yr period of +1.5 °C relatively to the reference period 1980 – 1999 and MAR forced by MIROC5 over the reference period |
| MARcan+2 | Anomalies between MAR forced by CanESM2 over a warmer 20-yr period of +2 °C relatively to the reference period 1980 – 1999 and MAR forced by MIROC5 over the reference period |

*Competing interests.* The authors declare that they have no conflict of interest.

*Acknowledgements.* Computational resources have been provided by the Consortium des Équipements de Calcul Intensif (CÉCI), funded by the Fonds de la Recherche Scientifique de Belgique (F.R.S.FNRS) under grant no. 2.5020.11 and the Tier-1 supercomputer (Zenobe) of the Fédération Wallonie Bruxelles infrastructure funded by the Walloon Region under the grant agreement no. 1117545.

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
