# Peer review of "Brief communication: Impact of the recent atmospheric circulation change in summer on the future surface mass balance of the Greenland ice sheet"

_The Cryosphere, 2018_

## Referee Comment (RC1) · Anonymous Referee #1 · 6 May 2018

**Overview**

This manuscript examines the inability of GCMs to reproduce recent high-latitude Northern Hemisphere circulation changes and the effect this has on model projections of future GrIS SMB. They force the MAR regional climate model with a number of different reanalysis and GCM boundary conditions, provided by the ERA-Interim reanalysis and three GCMs for the past climate and by the GCMs for the future climate. These sensitivity experiments ultimately show that GrIS SMB will be subject to much more significant future decreases if the recent (post-2000) shift toward negative sum-

mer NAO continues. They also show that GCMs that project temperature increases but do not capture recent circulation changes show a smaller decrease in SMB.

Overall, this work makes a useful contribution to our understanding of the effect of circulation changes on GrIS SMB and how well this is reproduced in GCMs. There are a few minor problems with the authors' characterization of recent circulation changes and the presentation of their methods and results. These issues and the recommended corrections are described in detail in the specific comments below.

Major comments

In the introduction, the authors partially attribute the recent increase in GrIS melt and mass loss to an increase in incoming solar radiation (p. 1–2). Similarly, in section 3.2 (p. 6), they state that "The current observed melt increase since the 2000's is partly due to the increase in downward shortwave radiation (SWD) caused by more frequent anticyclonic situations enhancing the melt-albedo feedback". In both cases, a single reference (Hofer et al., 2017) is provided. However, that study's claim – that decreasing summer cloud cover is driving the recent GrIS mass loss acceleration – is contradicted by a number of other works, which have demonstrated the important role of clouds and poleward moisture transport in providing melt energy during summer melt events. See, for example, see the Bennartz et al. 2013, Van Tricht et al. 2016, and Solomon et al. 2016 papers that show that clouds enhanced melt and/or reduced meltwater refreezing during recent major melting events. Also see the Neff et al. 2014, Bonne et al. 2015, Fausto et al. 2016, and Mattingly et al. 2016 papers, which together show that poleward moisture transport played a critical role in forcing the extreme July 2012 GrIS melt event, and that these types of moisture transport events have increased during the same 2000–2016 period discussed in this study.

The paper should be modified to more fairly reflect the breadth of the literature on this topic, noting that while one paper found a decreasing trend in summer cloud cover after 2000, most other studies on the topic have pointed to the key role played by

poleward transport of warm, moist air and the resultant cloud cover in forcing GrIS melt events. Including this information will also help align the characterization of recently observed circulation changes with the authors' statement that "both simulations forced by warmer reanalysis suggest a SWD decrease as well as in GCM-forced simulations with a warmer climate as a result of an increased cloud cover... This effect combined with a higher free atmosphere temperature explains then the increase in downward longwave radiation (LWD) in a warmer climate" (p. 6, lines 9–12).

Section 2.2.1, p. 3: More detail is needed here about the ERA-Interim atmospheric temperature forcing. Are the ERA-Interim atmospheric temperatures increased in a uniform manner at all vertical levels? Are they only increased near the surface? Or are they increased at 850 hPa, 700 hPa, 600 hPa, and 500 hPa, in a manner analogous to the temperature anomaly calculations for the GCMs (section 2.2.2)?

Figure 2 should be altered to include both positive and negative SMB anomalies (Fig. 2a) and differences (Fig. 2b) on a diverging color scale (like Figures S5 through S7 in the supplement). In addition to the areas of negative SMB anomalies / differences along the margins of the GrIS, the color scale is likely concealing areas of less intense positive anomalies / differences in the interior GrIS.

How is the statistical significance of anomalies calculated? (i.e. pg. 4, line 22; pg. 6, line 2)

The manner in which SMB anomalies in the experiments are discussed is confusing. On pg. 5 (lines 4–10), the SMB anomalies in the MARera2K+x experiments are described as having "significantly more negative" SMB anomalies and an "enhanced decrease in SMB" compared to the warmer reanalysis and GCM-forced experiments. However, in the Conclusions (pg. 7, lines 27–31), the SMB anomalies in the experiments with warming and a circulation change (i.e. the MARera2K+x experiments) are first described as having SMB anomalies that are "two times higher on average", then are described as having a "higher decrease in SMB". The language in the Results

and/or the Conclusions should be edited to be consistent, and to make the nature of the SMB anomalies clarified.

Minor comments

p. 1, l. 10: change "is similar" to "are similar" p. 1, l. 13: change "atmospheric conditions will persist" to "atmospheric conditions persist" p. 1, l. 20: change "have been attributed" to "has been attributed" p. 1, l. 21: misspelled "heighten" p. 2, l. 1: change "solar radiations" to "solar radiation" p. 2, l. 7: change "rises" to "raises" p. 2, l. 18: change "relatively" to "relative" p. 2, l. 19: change "consists in" to "consists of" p. 2, l. 20: change "radiations" to "radiation" and "precipitations" to "precipitation" p. 4, l. 8: change "Like with" to "As with" p. 7, l. 19: change "First experiments consisted in" to "The first experiments consisted of" Supplement: change "relatively" to "relative" throughout Table S2

References

Bennartz, R., M. D. Shupe, D. D. Turner, V. P. Walden, K. Steffen, C. J. Cox, M. S. Kulie, N. B. Miller, and C. Pettersen (2013), July 2012 Greenland melt extent enhanced by low-level liquid clouds, Nature, 496(7443), 83–86, doi:10.1038/nature12002.

Van Tricht, K., S. Lhermitte, J. Lenaerts, I. V. Gorodetskaya, T. S. L'Ecuyer, B. Noël, M. R. Van den Broeke, D. D. Turner, and N. P. Van Lipzig (2016), Clouds enhance Greenland ice sheet meltwater runoff, Nature Communications, 7, 10266, doi:10.1038/ncomms10266.

Solomon, A., M. D. Shupe, and N. B. Miller (2017), Cloud–atmospheric boundary layer–surface interactions on the Greenland ice sheet during the July 2012 extreme melt event, Journal of Climate, 30(9), 3237–3252, doi:10.1175/JCLI-D-16-0071.1.

Neff, W., G. P. Compo, F. Martin Ralph, and M. D. Shupe (2014), Continental heat anomalies and the extreme melting of the Greenland ice surface in 2012 and 1889, Journal of Geophysical Research: Atmospheres, 119(11), 6520–6536,

doi:10.1002/2014JD022602.

Bonne, J.-L., H. C. Steen-Larsen, C. Risi, M. Werner, H. Sodemann, J.-L. Lacour, X. Fettweis, G. Cesana, M. Delmotte, O. Cattani, P. Vallelonga, H. A. Kjær, C. Clerbaux, Á. E. Sveinbjörnsdóttir, and V. Masson-Delmotte (2015), The summer 2012 Greenland heat wave: In situ and remote sensing observations of water vapor isotopic composition during an atmospheric river event, Journal of Geophysical Research: Atmospheres, 120(7), 2970–2989, doi:10.1002/2014JD022602.

Fausto, R. S., D. van As, J. E. Box, W. Colgan, and P. L. Langen (2016a), Quantifying the surface energy fluxes in south Greenland during the 2012 high melt episodes using in-situ observations, Frontiers in Earth Science, 4, 82, doi:10.3389/feart.2016.00082.

Mattingly, K.S., C. A. Ramseyer, J. J. Rosen, T. L. Mote, and R. Muthyala (2016), Increasing water vapor transport to the Greenland Ice Sheet revealed using self-organizing maps, Geophysical Research Letters, 43(17), 9250–9258, doi:10.1002/2016GL070424.

---

## Referee Comment (RC2) · Anonymous Referee #2 · 26 May 2018

**Review of: "Brief communication: Impact of the recent atmospheric circulation change in summer on the future surface mass balance of the Greenland ice sheet", by *A. Delhasse et al.*, submitted to *The Cryosphere*.**

Since the 2000's, the Greenland ice sheet (GrIS) has experienced a circulation change toward more frequent anticyclonic conditions in summer, resulting in warmer near-surface conditions and increased melt. While regional climate models (RCM) forced by present-day climate re-analyses successfully reproduce this recent change, general circulation models (GCM) from the CMIP5 effort fail to do so, questioning their ability to provide reliable projections of the GrIS surface mass balance (SMB) if the ongoing circulation shift persists through the next decades. Using the regional climate model MAR forced by ERA-Interim re-analysis (1980-2016) and by RCP4.5 scenarios from three CMIP5 members (MIROC5, NorESM1 and CanESM2; 1980-2060), the authors investigate the impact of a persistent circulation shift (as currently observed) on GCM-forced projections of future GrIS SMB.

To that end, a set of sensitivity experiments consisting of MAR forced by perturbed ERA-Interim forcing (1980-2016), i.e. +1, +1.5, +2ºC hereafter called "warmer reanalysis", are conducted. Then the authors compare SMB anomalies derived from these "warmer reanalysis" experiments to MAR simulations forced by three GCMs over selected 20-year periods experiencing a similar temperature increase in the free atmosphere (+1, +1.5, +2ºC) relative to the reference period 1980-1999, i.e. before the onset of the recent circulation shift.

No significant difference in SMB can be found between the "warmer reanalysis" experiments for the reference period (1980-1999) and the GCM-forced simulations over the corresponding selected periods (+1, +1.5, +2ºC). For 2000-2016, i.e. including the recent circulation change, the "warmer reanalysis" experiments suggest a significant increase in meltwater runoff compared to the corresponding GCM-forced simulations, further resulting in two times lower SMB estimates. This highlights the importance of capturing current (and future) circulation change in GCM scenarios to provide reliable RCM-based projections of the GrIS SMB.

**General comments**

This is an original study based on a state-of-the-art, thoroughly evaluated climate model that will be of great interest to the cryospheric community. Additional clarifications and more detailed analysis of the results are necessary in places. The manuscript is sometimes confusing due to repetitions and lengthy sentences, which should be shortened/reformulated before acceptance. I deem that **minor revisions** are further required before publication in the Cryosphere. Hereunder, the authors can find suggestions listed as substantive, point and stylistic comments.

**Substantive Comments**

1. As mentioned by reviewer #1, additional information on how temperature perturbations are applied to the ERA-Interim forcing are necessary to better understand the results. Were the temperatures increased only at the surface or at each MAR atmospheric vertical level? This should be clearly mentioned. Section 2.1 should also explicitly state how many atmospheric vertical levels are used in these simulations.
2. Section 2.1 should also briefly discuss how the snow pack was initialized for the different sensitivity experiments. Is the initial state of the snow pack similar for each sensitivity experiment (MAR forced by ERA-Interim and GCM scenarios)?
3. In Section 2.2.2, the authors should explain in more detail why these three specific GCMs were selected. The authors should also clarify why the 20-yr periods experiencing +1, +1.5 and +2ºC are sometimes very different for the three GCMs, i.e. especially for CanESM2.

4. At P5 L3-5, the authors state that capturing the circulation change results in a massive runoff increase "nearly two times higher" relative to the reference period. This is an interesting result that is not further discussed. The authors should consider discussing the potential mechanisms driving this significant runoff increase. See also the corresponding point comment at P7 L4-6.

**Point Comments**

**P1 L4**: Add "North" before "Atlantic". **L8:** For consistency, replace "forced with" by "forced by". This comment holds for the whole manuscript. **L23**: The authors could add: "[…] snow grain metamorphism and further decreasing surface albedo […]".

**P2 L1**: The authors could add: "[…] in summer leads to longer exposure of bare ice at the GrIS margins […]". **L4-7**: The authors certainly mean that as GCMs fail to capture the current circulation change, the resulting recent melt increase modeled by RCMs forced by GCM "historical climate" is underestimated compared to observations. Could the authors clarify this and reformulate? **L21-27**: This paragraph should better be moved to Section 2.1. Section 2.2 could start at L27: "We performed two sets […]". Information about the number of atmospheric vertical levels and initialization of the snow pack could be briefly discussed in Section 2.1, see also substantive comments.

**P3** Sections 2.2.1 and 2.2.2 could be titled "ERA-Interim forcing" and "GCM forcing", respectively. **L18**: How are temperature in the free atmosphere estimated at 850-700 hPa when these pressure levels cross the surface topography of the GrIS interior?

**P4 L25-28**: I do not fully understand the analogy between SMB anomalies in Noël et al. (2014) and the present study. Could the authors clarify and reformulate? I also suggest: "These differences at the ice sheet margins **are similar to** SMB anomalies found […], who obtained insignificant impact […]".

**P5 L5**: I understand: "The **SMB** anomaly in MARera2k+1 […] more negative than the warmer reanalysis **over the reference period (MARera+1, resp. MARera+2) and the corresponding GCM-forced** future experiments (Table 2)", could the authors clarify? **L6-7**: Could the authors consider: "This suggests that capturing the recent circulation change simulated by warmer reanalysis in GCM-forced experiments would enhance the projected SMB decrease." Then at **L9**: "This is illustrated […] of +2ºC over 2000-2016 (Fig. 2b), i.e. including the recent circulation change, compared to the reference circulation over […]".

**P6 L9**: I read 3.7 W/m$^2$ in Table 2. The authors certainly mean "~4W/m$^2$". **L9-11**: The second part of this sentence is poorly written (i.e. after as well as), could the authors reformulate?

**P7 L1**: Table 2 shows that absorbed SWD is **more** than two times higher for 2000-2016 compared to the reference period. I suggest: "is more than two times". **L4-6**: As mentioned in the substantive comments, this is an interesting result which is unexploited. The authors should briefly elaborate on how increased melt lead to enhanced runoff, the authors could refer to Machguth et al. (2016). **L31**: Following my previous comment, melt is not a direct component of SMB. It is the runoff increase that drives the decrease in SMB.

**Stylistic suggestions**

**P1 L5**: Remove "in a warmer climate". **L6-9**: I would suggest to reformulate as follows: "We compare GrIS […] MAR forced by perturbed ERA-Interim reanalysis over 1980-2016, i.e. with a temperature increase of +1, +1.5, +2ºC relative to 1980-1999, to future […] forced by three GCMs over selected periods experiencing a similar temperature increase." **L11**: Remove "However,". **L18**: I would suggest: "multiple melting records have been broken […] ". **L19**: Replace "have been" by "can be". **L20:** Maybe "resulting from" instead of "gauged through". **L21**: Maybe "enhanced" instead of "heighten".

**P2 L3:** Remove "when they are" and add "climate" before reanalysis. **L7-8**: I would suggest: "This raises the question of how RCM-based projections of future GrIS SMB are affected by the GCM forcing if the recent shift to negative NAO phases in summer persists through the next decades." **L9:** Maybe: "we use the Modéle Atmosphérique Régional (MAR), especially developed for modeling the SMB of polar regions, to perform […] with perturbed ERA-Interim reanalysis (+1, +1.5, +2ºC) and three […]". **L18**: Relative to the version […]. **L19**: "consists of". **L20**: "radiation […] precipitation". **L23**: Replace "a global forcing dataset such as reanalysis" by "climate reanalyses". **L27:** Remove "only" and replace "in relation to" by "with respect to".

**P3** **L9:** "[…] lateral boundaries is conserved by estimating the specific humidity changes as a function of temperature increase". **L10-11:** Either "[…] with warmer […] atmospheric conditions" or "[…] with higher […] atmospheric temperature". **L6-13:** I suggest: "Therefore we compare anomalies of these GCM-forced […] and ~+2ºC to the corresponding GCM-forced […]. As for the sensitivity experiments […], the mean SMB anomalies associated with these GCMs […] x equals 1, 1.5 or 2 corresponds to +1ºC […] warming (Appendix […]). Contrary to ERA-Interim forced experiments, no humidity correction […] and the SSC are directly prescribed from RCP4.5 […]". The authors certainly mean "humidity correction" at L12?

**P4** **L16-19:** I would suggest: "[…] future SMB projections, we evaluate analogies between MAR forced by warmer reanalysis and by GCM future scenarios over periods experiencing a similar warmer climate. Figure 1 shows the difference in SMB anomalies between MARera+1 (resp. +2) and MARmir+1 (resp. +2, using MIROC5 as forcing) over 1980-1999 […]". **L28:** "Because GCMs fail […] ", **L29:** "similar SMB anomalies as MAR", **L31:** I guess the authors mean "a circulation change on the **projected** GrIS SMB".

**P5** **L3:** The authors could remove "in summer" as it is already suggested by "JJA temperature" at L4. **L13:** "As for SMB anomalies,".

**P7** **L12:** I would suggest: "The goal […] the impact of unresolved recent atmospheric circulation change in GCMs on RCM-based projections of future GrIS SMB." **L22:** "for which a similar temperature increase of". **L23:** "by these GCMs in the free atmosphere." **L24:** Replace "that the results are similar" by "similar results". **L28:** Maybe: "suggests that capturing the circulation change leads to SMB anomalies two times higher on average for a similar […]". **L32:** "The results […] suggest"

**P8** **L2-3:** "of examining whether GCMs can predict […] if so, evaluate […]".

**Figures and Tables**

**Table1:** For consistency, replace 1 ± 0.39 by 1.00 ± 0.39 in the second row of the second column.

**Table2:** The authors should consider to explicitly mention MARera and MARera2k instead of/in addition to ERA-Interim in column 3 and 4.

**Figure1:** To improve readability, could the hatches be displayed in a darker color e.g. grey?

**Figure2:** As this figure also shows anomalies, a red-to-blue color scale centered on 0 should be used. As for Figure1, hatches could also be displayed in grey for better visibility.

**Appendix A1 and A2:** For consistency, replace "forced with" by "forced by". The same applies to the two similar tables in the Supplementary Material.

**Additional reference and DOI**

Machguth et al. (2016) DOI: 10.1038/NCLIMATE2899

---

## Author Response (AR2)

Dear Editor, Dear Marco,

First we would like to thank reviewers for their relevant comments which have helped us to improve our manuscript.

Responses to each individual reviewer have been posted on TCD.

The main changes in respect to the original version are:

- The introduction has been improved by mentioning effect of clouds on surface melting, using the various references suggested by the reviewer #1;

- Melt anomalies are now discussed in the results section, as suggested by reviewer #2;

- Hatched zone representing non-significant differences in the different figures are now black hatched to increase visibility.

All the minor corrections and improvements suggested by reviewers have been taken into account in the revised version of the manuscript.

All the best,

Alison Delhasse

We first would like to thank the reviewer comments which will help to improve our manuscript.

Overview This manuscript examines the inability of GCMs to reproduce recent high-latitude Northern Hemisphere circulation changes and the effect this has on model projections of future GrIS SMB. They force the MAR regional climate model with a number of different reanalysis and GCM boundary conditions, provided by the ERA-Interim reanalysis and three GCMs for the past climate and by the GCMs for the future climate. These sensitivity experiments ultimately show that GrIS SMB will be subject to much more significant future decreases if the recent (post-2000) shift toward negative sum TCD Interactive comment Printer-friendly version Discussion paper mer NAO continues. They also show that GCMs that project temperature increases but do not capture recent circulation changes show a smaller decrease in SMB. Overall, this work makes a useful contribution to our understanding of the effect of circulation changes on GrIS SMB and how well this is reproduced in GCMs. There are a few minor problems with the authors' characterization of recent circulation changes and the presentation of their methods and results. These issues and the recommended corrections are described in detail in the specific comments below.

Major comments

In the introduction, the authors partially attribute the recent increase in GrIS melt and mass loss to an increase in incoming solar radiation (p. 1–2). Similarly, in section 3.2 (p. 6), they state that "The current observed melt increase since the 2000's is partly due to the increase in downward shortwave radiation (SWD) caused by more frequent anticyclonic situations enhancing the melt-albedo feedback". In both cases, a single reference (Hofer et al., 2017) is provided. However, that study's claim – that decreasing summer cloud cover is driving the recent GrIS mass loss acceleration – is contradicted by a number of other works, which have demonstrated the important role of clouds and poleward moisture transport in providing melt energy during summer melt events. See, for example, see the Bennartz et al. 2013, Van Tricht et al. 2016, and Solomon et al. 2016 papers that show that clouds enhanced melt and/or reduced meltwater refreezing during recent major melting events. Also see the Neff et al. 2014, Bonne et al. 2015, Fausto et al. 2016, and Mattingly et al. 2016 papers, which together show that poleward moisture transport played a critical role in forcing the extreme July 2012 GrIS melt event, and that these types of moisture transport events have increased during the same 2000–2016 period discussed in this study.

The paper should be modified to more fairly reflect the breadth of the literature on this topic, noting that while one paper found a decreasing trend in summer cloud cover after 2000, most other studies on the topic have pointed to the key role played by poleward transport of warm, moist air and the resultant cloud cover in forcing GrIS melt events. Including this information will also help align the characterization of recently observed circulation changes with the authors' statement that "both simulations forced by warmer reanalysis suggest a SWD decrease as well as in GCM-forced simulations with a warmer climate as a result of an increased cloud cover. This effect combined with a higher free atmosphere temperature explains then the increase in downward longwave radiation (LWD) in a warmer climate" (p. 6, lines 9–12).

These relevant remarks will be taken into account to improve our introduction where the antagonist role of clouds over the ablation zone (where they rather cold the climate) and over the accumulation

zone (where they rather warm the climate) will be discussed in more details, as well as our discussion of results (Section x.y).

Section 2.2.1, p. 3: More detail is needed here about the ERA-Interim atmospheric temperature forcing. Are the ERA-Interim atmospheric temperatures increased in a uniform manner at all vertical levels? Are they only increased near the surface? Or are they increased at 850 hPa, 700 hPa, 600 hPa, and 500 hPa, in a manner analogous to the temperature anomaly calculations for the GCMs (section 2.2.2)?

More details should be given in the sentence p3, line 5-6. We suggest then to reformulate our sentence " Then, we performed sensitivity experiments in which ERA-Interim atmospheric temperatures were increased by respectively +1 °C, +1.5 °C and +2 °C at the MAR atmospheric lateral boundaries (hereafter referred to as warmer reanalysis)."
to
"Then, we performed sensitivity experiments in which ERA-Interim atmospheric temperatures were increased by respectively +1 °C, +1.5 °C and +2 °C at each of the 24 vertical sigma level of the MAR atmospheric lateral boundaries (hereafter referred to as warmer reanalysis)."

Figure 2 should be altered to include both positive and negative SMB anomalies (Fig. 2a) and differences (Fig. 2b) on a diverging color scale (like Figures S5 through S7 in the supplement). In addition to the areas of negative SMB anomalies / differences along the margins of the GrIS, the color scale is likely concealing areas of less intense positive anomalies / differences in the interior GrIS.

Adapted figure:

[Figure]

How is the statistical significance of anomalies calculated? (i.e. pg. 4, line 22; pg. 6, line 2)

Anomalies and/or differences are significant if they are greater than the standard deviation (i.e. interanual variability of the annual SMB) of the simulation of MAR forced by unaltered ERA-Interim over 1980-1999 as explained p4, line 22 and in the legends of both figures.

The manner in which SMB anomalies in the experiments are discussed is confusing. On pg. 5 (lines 4–10), the SMB anomalies in the MARera2K+x experiments are described as having "significantly more negative" SMB anomalies and an "enhanced decrease in SMB" compared to the warmer reanalysis and GCM-forced experiments. However, in the Conclusions (pg. 7, lines 27–31), the SMB anomalies in the experiments with warming and a circulation change (i.e. the MARera2K+x experiments) are first described as having SMB anomalies that are "two times higher on average", then are described as having a "higher decrease in SMB". The language in the Results and/or the Conclusions should be edited to be consistent, and to make the nature of the SMB anomalies clarified.

To be consistent, we will change " ...  two times higher on average ..." in " ...  two times more negative on average..." in p7, line 27.

Minor comments

p. 1, l. 10: change "is similar" to "are similar"
p. 1, l. 13: change "atmospheric conditions will persist" to "atmospheric conditions persist"
p. 1, l. 20: change "have been attributed" to "has been attributed"
p. 1, l. 21: misspelled "heighten"
p. 2, l. 1: change "solar radiations" to "solar radiation"
p. 2, l. 7: change "rises" to "raises"
p. 2, l. 18: change "relatively" to "relative"
p. 2, l. 19: change "consists in" to "consists of"
p. 2, l. 20: change "radiations" to "radiation" and "precipitations" to "precipitation"
p. 4, l. 8: change "Like with" to "As with"
p. 7, l. 19: change "First experiments consisted in" to "The first experiments consisted of"
Supplement: change "relatively" to "relative" throughout Table S2

Ok, thanks. All of these will be taken into account in the revised version of our paper.

We first would like to thank the reviewer for his comments which will help to improve our manuscript.

*Substantive Comments*

1. As mentioned by reviewer #1, additional information on how temperature perturbations are applied to the ERA-Interim forcing are necessary to better understand the results. Were the temperatures increased only at the surface or at each MAR atmospheric vertical level? This should be clearly mentioned. Section 2.1 should also explicitly state how many atmospheric vertical levels are used in these simulations.

As explained in the responds to reviewer #1, more details will be given in the sentence p3, line 5-6. We suggest then to reformulate our sentence "*Then, we performed sensitivity experiments in which ERA-Interim atmospheric temperatures were increased by respectively +1 °C, +1.5 °C and +2 °C at the MAR atmospheric lateral boundaries (hereafter referred to as warmer reanalysis)."*
to
"*Then, we performed sensitivity experiments in which ERA-Interim atmospheric temperatures were increased by respectively +1 °C, +1.5 °C and +2 °C at each of the 24 vertical sigma level of the MAR atmospheric lateral boundaries (hereafter referred to as warmer reanalysis)."*

2. Section 2.1 should also briefly discuss how the snow pack was initialized for the different sensitivity experiments. Is the initial state of the snow pack similar for each sensitivity experiment (MAR forced by ERA-Interim and GCM scenarios)?

The reference simulations have been initialized with snowpacks from previous MAR simulations (forced by ERA and by the 3 GCMs) and started in 1970 to give time to MAR to be independent of the initial snow condition. To remove the dependence of the snowpack initialization in the ERA-Interim forced sensitivity experiments, we have started these simulations in 1970 with warmer ERA-40.

3. In Section 2.2.2, the authors should explain in more detail why these three specific GCMs were selected. The authors should also clarify why the 20-yr periods experiencing +1, +1.5 and +2 °C are sometimes very different for the three GCMs, i.e. especially for CanESM2.

As explain p3 l 21-22, the three selected GCMs are the best representing the general circulation at 500 hPa (impacting the precipitation amount and pattern simulated by MAR) and the JJA (June-July-August) temperature at 700 hPa (impacted the melt amount simulated by MAR) over Greenland compared to ERA-Interim over 1980 – 1999.We refer to Fettweis et al. (2013) for more details in this choice of GCMs

The 20-yr periods experiencing +1, +1.5 and +2 °C are very different following the used GCM because there is offset in the warming projected by each GCM: For instance, CanESM2 projects a faster warming notably due to the melting of the Arctic sea ice with respect to the other GCMs. Again, this is also well shown and discussed in Fettweis et al. (2013).

4. At P5 L3-5, the authors state that capturing the circulation change results in a massive runoff increase "nearly two times higher" relative to the reference period. This is an interesting result that is not further discussed. The authors should consider discussing the potential mechanisms driving this significant runoff increase. See also the corresponding point comment at P7 L4-6.

See point comment at P7 L4-6.

*Point Comments*

P1      L4: Add "North" before "Atlantic".
L8: For consistency, replace "forced with" by "forced by". This comment holds for the whole manuscript.
L23: The authors could add: "[…] snow grain metamorphism and further decreasing surface albedo […]".

Ok, thanks. All of these will be taken into account in the revised version of our paper.

P2      L1: The authors could add: "[…] in summer leads to longer exposure of bare ice at the GrIS margins […]".

OK, thanks.

L4-7: The authors certainly mean that as GCMs fail to capture the current circulation change, the resulting recent melt increase modeled by RCMs forced by GCM "historical climate" is underestimated compared to observations. Could the authors clarify this and reformulate?

GCMs do not simulate any circulation change for both the historical scenario (prior to 2006) and RCPs scenarios, so that the melt increase observed since the 2000's is underestimated when RCM's are forced by these GCM as Fettweis et al. (2013a) showed that 70% of the recent melt increase is explained by the NAO shift. We therefore propose to reformulate L4-7 (p2):

"*Such an amplification in surface melt is well represented by Regional Climate Models (RCMs) when they are forced by reanalysis (Ettema et al., 2009; Fettweis et al., 2011, 2017; Noël et al., 2015, 2018). However, as General Circulation Models (GCMs) do not presently predict any circulation changes (Belleflamme et al., 2012; Fettweis et al., 2013b), the melt increase currently observed is underestimated when RCMs are forced by GCM scenarios (e.g., Fettweis et al., 2011, 2013b; Rae et al., 2012).*"

to

"*Such an amplification in surface melt is well represented by Regional Climate Models (RCMs) when they are forced by reanalysis which capture the current circulation change (Ettema et al., 2009; Fettweis et al., 2011, 2017; Noël et al., 2015, 2018). However, as General Circulation Models (GCMs) do not presently predict any circulation changes (Belleflamme et al., 2012; Fettweis et al., 2013b), the melt increase currently observed is underestimated when RCMs are forced by GCM scenarios starting from 2000's (e.g., Fettweis et al., 2011, 2013b; Rae et al., 2012).*"

L21-27: This paragraph should better be moved to Section 2.1. Section 2.2 could start at L27: "We performed two sets […]".

OK, thanks.

Information about the number of atmospheric vertical levels and initialization of the snow pack could be briefly discussed in Section 2.1, see also substantive comments.

See Substantive Comment 1 for the number of atmospheric vertical levels (i.e, 24 levels) and Substantive Comment 2 for the initialization of the snowpack.

P3    Sections 2.2.1 and 2.2.2 could be titled "ERA-Interim forcing" and "GCM forcing", respectively.

OK, thanks.

L18: How are temperature in the free atmosphere estimated at 850-700 hPa when these pressure levels cross the surface topography of the GrIS interior?

If the topography is higher than the altitude of the level pressure, the pixel is not taken into account for the average temperature.

P4    L25-28: I do not fully understand the analogy between SMB anomalies in Noël et al. (2014) and the present study. Could the authors clarify and reformulate?
I also suggest: "These differences at the ice sheet margins are similar to SMB anomalies found […], who obtained insignificant impact […]".

Although we made the analogy between forcing MAR by reanalyses warmer by 1, 1.5 and 2°C over the 1980-1999 period and by GCMs over a climate warmer by 1, 1.5 and 2°C compared to their reference climate over 1980-1999, experiments based on warmer reanalyses differ from corresponding experiments based on GCM because sea surface conditions (SSC, namely SIC and SST) remains unchanged in the warmer reanalysis forced sensitivity experiments but correspond to a warmer climate in the GCM forced simulations. SSC in MARera+x are thus representative of a colder ocean (more SIC and less SST) than the SSC from MARnor, MARcan and MARmir experiments. On the other hand, Noël et al. (2014) evaluated the influence of warmer SSC on the Greenland SMB by increasing (resp. decreasing) SST (resp. SIC) of ERA-Interim. Differences at the ice sheet margins (Fig. 1) are similar to the SMB anomalies found by Nöel et al (2014). We therefore assume that weak anomalies between MARera+x and MARmir (or MARnor or MARcan) result from the SSC unchanged in MARera+x.

We will modify L25-28:

*"However, these differences on the ice sheet margins correspond to the same anomalies found by Noël et al. (2014) who evaluated the (not significant) impact of warmer SSC on the current SMB. Therefore, we can reasonably assume that these differences in anomalies mainly result from SSC not modified in experiments based on warmer reanalysis compared to GCM-forced simulations using future SSC."*

to

*"These weak differences are caused by the sea surface conditions (SSC) which were not modified in experiments based on warmer reanalyses while the GCM-forced simulations use future SSC and corresponds to what found by Noël et al. (2014) who showed that same unsignificant anomalies warmer SSC."*

P5    L5: I understand: "The SMB anomaly in MARera2k+1 […] more negative than the warmer reanalysis over the reference period (MARera+1, resp. MARera+2) and the corresponding GCM-forced future experiments (Table 2)", could the authors clarify?

The SMB anomaly MARera2k+1 (resp. MARera2k+1.5) is significantly more negative than warmer reanalysis experiments and GCM-forced future experiments relative to a climate warmer by +1.5 °C (resp. +2 °C).

L6-7: Could the authors consider: "This suggests that capturing the recent circulation change simulated by warmer reanalysis in GCM-forced experiments would enhance the

projected SMB decrease." Then at L9: "This is illustrated […] of +2 ºC over 2000-2016 (Fig. 2b), i.e. including the recent circulation change, compared to the reference circulation over […]".

OK, thanks.

P6     L9: I read 3.7 W/m² in Table 2. The authors certainly mean "~4W/m2".

OK, thanks.

L9-11: The second part of this sentence is poorly written (i.e. after as well as), could the authors reformulate?

We will rewrite

*"However both simulations forced by warmer reanalysis suggest a SWD decrease as well as in GCM-forced simulations with a warmer climate as a result of an increased cloud cover (Franco et al., 2013)."*

to

*"However both simulations forced by warmer reanalysis as well as in GCM-forced simulations with a warmer climate suggest a SWD decrease as a result of an increased cloud cover (Franco et al.,2013)"*

P7     L1: Table 2 shows that absorbed SWD is more than two times higher for 2000-2016 compared to the reference period. I suggest: "is more than two times".

OK, thanks.

L4-6: As mentioned in the substantive comments, this is an interesting result which is unexploited. The authors should briefly elaborate on how increased melt lead to enhanced runoff, the authors could refer to Machguth et al. (2016).

We have calculated same anomalies than runoff for the production of meltwater (ME):

| Anomaly | Temperature increase (°C) | MARera+x | MARera2k+x | Mean 3 models |
|---------|---------------------------|----------|------------|---------------|
| Annual mean SMB (Gt) | +0 | 0 | **-205** | |
| | +1 | -84 | **-326** | -118 |
| | +1,5 | -146 | **-408** | -164 |
| | +2 | **-206** | -492 | **-197** |
| Annual mean RU (Gt) | +0 | 0 | 211 | |
| | +1 | **142** | 393 | 141 |
| | +1,5 | **236** | 508 | 215 |
| | +2 | **328** | 626 | 283 |
| Annual mean ME (Gt) | +0°C | **0** | 195 | |
| | +1°C | **133** | 352 | 135 |
| | +1,5°C | **210** | 440 | 203 |
| | +2°C | **291** | 534 | 261 |

And we suggest to modify the flowing sentences by adding some details (in blue) in paragraph starting P5 L12 and ending P6 L13:

"As runoff (RU) and snowfall (SF) mainly drive the GrIS SMB (Box et al., 2004), we discuss in the following the anomalies relative to these two components only. Like for SMB anomalies, RU and SF anomalies are computed as differences between the corresponding mean value for a given experiment and the mean value for the reference period using the unaltered large-scale forcing (Table 2). Even though non-significant, an increase in SF is observed for all experiments associated with temperature rising in response to a higher air capacity for holding water vapor (Fettweis et al., 2013a). Moreover, mean RU anomalies increase with the temperature rising in all warming experiments, most significantly for the experiments using warmer reanalysis over 2000 – 2016 when the circulation change has occurred. It can thus be concluded that runoff is mainly responsible for the SMB discrepancies between the different sensitivity experiments in a warmer climate. Melt (ME) is also amplifying as RU with the circulation change. However, RU anomalies are systematically higher than ME anomalies which means RU increase more than ME. It can be explain by two factors (Machguth et al., 2016): (1) there is less pore place available for meltwater storage in warmer firn and, (2) bare ice area (in the ablation zone) is larger in warmer climate, so there is less meltwater storage which amplifies the runoff increase. The future decrease of the ice sheet meltwater capacity retention was notably shown by Van Angelen et al. (2013).

Due to the enhanced positive melt-albedo feedback since the 2000's, SWD absorbed by the surface is two times higher in simulations with warmer reanalysis over 2000 – 2016 than over the reference period. Due to a lower albedo, the surface absorbs more energy, amplifying the melt increase which further decreases the albedo, potentially reaccelerating melting in addition to a decrease of the ice sheet capacity to refreeze meltwater. This positive feedback triggered by more frequent anticyclonic summer situations over Greenland causes a runoff increase nearly two times higher in simulations over 2000 – 2016 than in the simulations over the reference period, i.e. before the circulation change."

L31: Following my previous comment, melt is not a direct component of SMB. It is the runoff increase that drives the decrease in SMB.

We will rewrite :
"*As a result, the melt increase is enhanced and is responsible for the higher decrease in SMB.*"

to

"*As a result, the runoff increase is enhanced and is responsible for the higher decrease in SMB.*"

*Stylistic suggestions*

**P1**     L5: Remove "in a warmer climate".
L6--9: I would suggest to reformulate as follows: "We compare GrIS […] MAR forced by perturbed ERA--Interim reanalysis over 1980--2016, i.e. with a temperature increase of +1, +1.5, +2 °C relative to 1980--1999, to future […] forced by three GCMs over selected periods experiencing a similar temperature increase."
L11: Remove "However,"
L18: I would suggest: "multiple melting records have been broken [...] "
L19: Replace "have been" by "can be"
L20: Maybe "resulting from" instead of "gauged through".
L21: Maybe "enhanced" instead of "heighten".

**P2**     L3: Remove "when they are" and add "climate" before reanalysis.

L7–8: I would suggest: "This raises the question of how RCM–based projections of future GrIS SMB are affected by the GCM forcing if the recent shift to negative NAO phases in summer persists through the next decades."

L9: Maybe: "we use the Modéle Atmosphérique Régional (MAR), especially developed for modeling the SMB of polar regions, to perform [...] with perturbed ERA–Interim reanalysis (+1, +1.5,+2 °C) and three [...]".

L18:     Relative to the version [...].

L19: "consist of".

L20: "radiation [...] precipitation".

L23: Replace "a global forcing dataset such as reanalysis" by "climate reanalyses".

L27: Remove "only" and replace "in relation to" by "with respect to".

**P3**     L9: "[...] lateral boundaries is conserved by estimating the specific humidity changes as a function of temperature increase".

L10–11: Either "[...] with warmer [...] atmospheric conditions" or "[...] with higher [...] atmospheric temperature".

L6–13: I suggest: "Therefore we compare anomalies of these GCM–forced [...] and ~+2 °C to the corresponding   GCM–forced [...]. As for the sensitivity experiments [...], the mean SMB anomalies  associated with these GCMs [...] x equals 1, 1.5 or 2 corresponds to +1 °C [...] warming (Appendix    [...]). Contrary to ERA–Interim forced experiments, no humidity correction [...] and the SSC are directly prescribed from RCP4.5 [...]". The authors certainly mean humidity correction" at L12?

**P4**     L16–19: I would suggest: "[...] future SMB projections, we evaluate analogies between MAR forced by warmer reanalysis and by GCM future scenarios over periods experiencing a similar warmer climate. Figure 1 shows the difference in SMB anomalies between MARera+1 (resp. +2) and MARmir+1 (resp.+2,using MIROC5 as forcing) over 1980–1999 [...]".

L28: "Because GCMs fail [...]"

L29:     "similar SMB anomalies as MAR",

L31:     I guess the authors mean "a circulation change on the projected GrIS SMB".

**P5**      L3: The authors could remove "in summer" as it is already suggested by "JJA temperature" at L4.

L13: "As for SMB anomalies,"

**P7**     L12: I would suggest: "The goal [...] the impact of unresolved recent atmospheric circulation change in GCMs on RCM–based projections of future GrIS SMB."

L22: "for which a similar temperature increase of".

L23: "by these GCMs in the free atmosphere."

L24: Replace "that the results are similar "by "similar results".

L28: Maybe: "suggests that capturing the circulation change leads to SMB anomalies two times higher on average for a similar [...]".

L32: "The results […] suggest"

**P8**     L2–3: "of examining whether GCMs can predict [...] if so, evaluate [...]".

Ok, thanks. All of these will be taken into account in the revised version of our paper.

_Figures and Tables_

**Table1**: For consistency, replace 1± 0.39 by 1.00 ± 0.39 in the second row of the second column.
**Table2**: The authors should  consider to explicitly mention MARera and MARera2k instead of/in addition to ERA-Interim in column 3 and 4.
**Figure1**: To improve readability, could the hatches be displayed in a darker color e.g. grey?
**Figure2**:  As this figure also shows anomalies, a red-to-blue  color scale centered on 0 should be used.As for Figure1,  hatches could also be displayed in grey for better visibility.

**Appendix  A1 and A2**: For consistency, replace "forced with" by "forced by". The same applies to the two similar tables in the Supplementary Material.

Ok, thanks. All of these will be taken into account in the revised version of our paper.

References

[revised manuscript text omitted]